# No association between cataract surgery and mitochondrial DNA damage with age-related macular degeneration in human donor eyes

**Karen R. Armbrust**[1], **Pabalu P. Karunadharma**[1], **Marcia R. Terluk**[1], **Rebecca J. Kapphahn**[1], **Timothy W. Olsen**[1,2], **Deborah A. Ferrington**[1], **Sandra R. Montezuma**[1]*

**1** Department of Ophthalmology and Visual Neurosciences, University of Minnesota, Minneapolis, Minnesota, United States of America, **2** Department of Ophthalmology, Mayo Clinic, Rochester, Minnesota, United States of America

* smontezu@umn.edu

## Abstract

### Purpose

To determine whether age-related macular degeneration (AMD) severity or the frequency of retinal pigment epithelium mitochondrial DNA lesions differ in human donor eyes that have undergone cataract surgery compared to phakic eyes.

### Methods

Eyes from human donors aged $\geq$ 55 years were obtained from the Minnesota Lions Eye Bank. Cataract surgery status was obtained from history provided to Eye Bank personnel by family members at the time of tissue procurement. Donor eyes were graded for AMD severity using the Minnesota Grading System. Quantitative PCR was performed on DNA isolated from macular punches of retinal pigment epithelium to quantitate the frequency of mitochondrial DNA lesions in the donor tissue. Univariable and multivariable analyses were performed to evaluate for associations between (1) cataract surgery and AMD severity and (2) cataract surgery and mitochondrial DNA lesion frequency.

### Results

A total of 157 subjects qualified for study inclusion. Multivariable analysis with age, sex, smoking status, and cataract surgery status showed that only age was associated with AMD grade. Multivariable analysis with age, sex, smoking status, and cataract surgery status showed that none of these factors were associated with retinal pigment epithelium mitochondrial DNA lesion frequency.

### Conclusions

In this study of human donor eyes, neither retinal pigment epithelium mitochondrial DNA damage nor the stage of AMD severity are independently associated with cataract surgery after adjusting for other AMD risk factors. These new pathologic and molecular findings

**Data Availability Statement:** All relevant data are within the manuscript and its Supporting Information files.

**Funding:** Funding was provided by the Minnesota Lions Vision Foundation, the Elaine and Robert Larson Endowed Vision Research Chair and the Helen Lindsay Family Foundation (DAF), the Knobloch Chair Professorship (SRM), an ARVO travel award (KRA), and the University of Minnesota Harry Friedman resident research award (KRA). The funders had no role in study design, data collection and analysis, decision to publish, or preparation of the manuscript.

**Competing interests:** I have read the journal's policy and the authors of this manuscript have the following competing interests: TWO: Equity owner – iMacular Regeneration LLC. DAF: Scientific advisory board member – Vinci Pharmaceuticals. The authors have no proprietary or commercial interest in any materials discussed in this article. This does not alter our adherence to PLOS ONE policies on sharing data and materials.

provide evidence against a relationship between cataract surgery and AMD progression and support the idea that cataract surgery is safe in the setting of AMD.

## Introduction

Since sunlight exposure [1–3] and inflammation [4, 5] represent proposed risk factors for age-related macular degeneration (AMD) pathogenesis, cataract surgery has a theoretic potential to accelerate AMD disease progression. Retrospective studies show that good peri-operative control of exudative AMD is feasible [6, 7]; however, the possibility remains that increased light exposure during and after removal of the cataractous lens [8] and/or surgically-induced inflammation causes retinal and retinal pigment epithelium (RPE) damage that may worsen AMD.

Some population-based studies show an association between cataract surgery and progression of AMD. For instance, the Beaver Dam Eye Study [9] and the Blue Mountains Eye Study [10] both show an association between cataract surgery and more advanced AMD. Similarly, recent analysis of the Singapore Malay and Indian Eye Studies shows that eyes that underwent cataract surgery in these cohorts had a higher incidence of advanced AMD than unoperated eyes after adjusting for age, sex, ethnicity, and drusen/retinal pigmentation abnormalities at baseline [11]. On the other hand, an association between cataract surgery and more advanced AMD is not seen in the ANCHOR [12], MARINA, and AREDS2 [13] studies. The most recent Cochrane review on this topic determined that the available data are insufficient to make a conclusive statement about the long-term safety and effectiveness of cataract surgery in the setting of AMD [14].

We hypothesized that if cataract surgery worsens AMD, human donor eyes with a history of cataract surgery would have more advanced AMD and/or more retinal pigment epithelium (RPE) mitochondrial (mt) DNA damage. The link between mtDNA damage and AMD is supported by a prior study that showed decreased number and size of RPE mitochondria as well as loss of RPE mt matrix density and cristae in AMD [15]. Although these changes also occurred in normal aging, the changes were accelerated in AMD. Additional studies showed that human donor eyes with advanced AMD have more RPE mtDNA damage than eyes of age-matched controls [16–18]. In the current study, we evaluated whether AMD severity (based on the Minnesota Grading System (MGS) [19, 20]) and RPE mtDNA lesion frequency differ in human donor eyes with and without cataract surgery, after adjusting for other potential AMD risk factors. Inclusion of mtDNA damage analysis from donor eyes provides new information about features of the RPE that are not accessible in living patients due to the invasive nature of measuring damage in the RPE. Therefore, this study extends information provided by previous clinical studies by including a biochemical measure of mtDNA damage in the cell type affected by AMD.

## Materials and methods

This experimental study utilized human donor eyes procured during 2006–2015 from the Minnesota Lions Eye Bank (renamed the Lions Gift of Sight in 2018, St. Paul, MN, URL: https://lionsgiftofsight.umn.edu), which provided de-identified specimens and medical history. The University of Minnesota Institutional Review Board (IRB)/Ethics Committee ruled that approval was not required for this study. The Eye Bank protocols are in accordance with University of Minnesota ethical standards and adhere to the tenets of the Declaration of

Helsinki. Reports from the Eye Bank specified demographics of the donors including age, gender, race, and a family report of a limited medical and ocular history. Cataract surgery status and smoking history were determined from information obtained by Eye Bank personnel from family members at the time of tissue procurement. Exclusion criteria for this study were age < 55 years or unclear cataract surgery status.

## AMD grading

The evaluation and classification of the donor's stage of AMD was performed by a clinician (TWO and SRM) from stereoscopic fundus photographs of the RPE using criteria established by the Minnesota Grading System (MGS) for eye bank eyes [19, 20]. The MGS is a four-level ordinal scale. MGS 1 corresponds to the control group with no clinically observable AMD. MGS 2, 3, 4 are early, intermediate, and advanced stages of AMD, respectively. Advanced AMD (MGS 4) corresponds to either non-exudative AMD with central geographic atrophy or exudative AMD with choroidal neovascularization.

## Mitochondrial DNA analysis

Quantitative PCR using primers that recognized specific regions of the mitochondrial genome was performed on DNA isolated from 5 mm macular punches of retinal pigment epithelium (RPE) to assess the extent of mitochondrial DNA damage [17, 18]. This assay uses mitochondrial-specific primers that preferentially amplify mtDNA from total DNA isolated from RPE. The advantage of this method is that mitochondrial purification is not required. Genomic regions without DNA damage amplify more efficiently with quantitative PCR than regions with DNA damage [21], so this assay allows quantitative determination of mtDNA damage based on PCR amplification efficiency. As previously described [22], this study combined mtDNA lesion frequency data from two separate studies by normalizing mtDNA lesion frequency for each subject to the mean mtDNA lesion frequency of age-matched controls (MGS1) in the corresponding study. We utilized the RPE mtDNA data from this prior study with new cataract surgery data to investigate the potential relationship between RPE mtDNA damage and cataract surgery.

## Statistical analyses

Descriptive statistics to summarize data and histogram plots were constructed in Microsoft Excel. Boxplots were constructed with the ggplot2 package in R statistical computing [23]. Wilcoxon rank sum tests and two-tailed two-sample Student's t-tests for univariable analysis and multiple regression analyses using R version 4.0.3 [24] were used to test for associations between the dependent variables (MGS grade and mtDNA lesion frequency) and independent variables (cataract surgery status and known risk factors for AMD including age, female gender, and smoking status). Equivalence hypothesis testing was performed with the TOSTER package [25]. All variables in all multivariable analyses were verified to have a variance inflation factor < 5. The MGS grade for most subjects was the same in both eyes; in subjects with two different MGS grades, the higher MGS grade was used for statistical analysis. Since the majority of subjects either had not had cataract surgery or had bilateral cataract surgery, the statistical analyses excluded those with unilateral cataract surgery and cataract surgery was changed to a dichotomous person variable (0 = no cataract surgery, 1 = bilateral cataract surgery). Race was not included in the analyses because the entire study population was Caucasian. $P < 0.05$ was predetermined to meet statistical significance.

## Results

A total of 157 subjects met study criteria. Table 1 presents the demographic and cataract surgery status data for the study population as well as information about MGS subgroups. The mean (standard deviation, range) age of the study subjects was 79 (9.0, 58–101) years. The study population consisted of 57% females and 52% smokers.

The cataract surgery status for the majority of study subjects was the same for both eyes, with only 3% of subjects having unilateral cataract surgery. Four of the five subjects with a history of cataract surgery in only one eye had the same MGS grade in both eyes. In the remaining subject the MGS grade was higher in the nonsurgical eye than in the surgical eye. Therefore, in all five subjects with unilateral cataract surgery, MGS grade was not higher in the surgical eye. Although the number of individuals with unilateral cataract surgery is low, these results support the idea that cataract surgery does not contribute to advancing AMD disease severity.

Univariable analysis showed different distributions in MGS grade in subjects without cataract surgery compared to those with bilateral cataract surgery (Fig 1A), and this difference is statistically significant (Wilcoxon rank sum, $P = 0.043$). However, there also is a significant association between age and cataract surgery status (Fig 1B, Student's t-test, $P < 0.001$); the mean age (standard deviation) is 74.7 (8.5) years in the group without cataract surgery and 83.3 (7.4) years in the group with bilateral cataract surgery. Expanding the analysis to a multivariable ordinal logistic regression model with MGS grade as the dependent variable and cataract surgery status, age, gender, and smoking as the independent variables shows that only age has a significant association with MGS grade ($P < 0.001$, Table 2).

Our hypothesis that cataract surgery promotes AMD progression also predicts that the more advanced stages of AMD (MGS 3 and 4) would be associated with cataract surgery. However, limiting the multivariable analysis by comparing intermediate/late AMD (MGS 3 and 4) to controls (MGS 1) also showed that only age has a significant association with MGS grade ($P < 0.001$, Table 3). Thus, after adjusting for age, we conclude that the donor eye history of cataract surgery is not associated with MGS grade.

**Table 1. Demographic characteristics and cataract surgery status of the study population, _N_ = 157.**

| Variable | N (%) | MGS 1 (control) | MGS 2 (early AMD) | MGS 3 and 4 (intermediate/late AMD) |
|---|---|---|---|---|
| Age (yrs) | | | | |
| • 55–64 | 13 (8) | 9 | 2 | 2 |
| • 65–74 | 33 (21) | 15 | 12 | 6 |
| • 75–84 | 62 (39) | 22 | 14 | 26 |
| • 85–94 | 44 (28) | 8 | 12 | 24 |
| • 95+ | 5 (3) | 0 | 0 | 5 |
| Gender | | | | |
| • Male | 68 (43) | 29 | 19 | 20 |
| • Female | 89 (57) | 25 | 21 | 43 |
| Smoking status | | | | |
| • Smoker | 81 (52) | 33 | 18 | 30 |
| • Nonsmoker | 76 (48) | 21 | 22 | 33 |
| Cataract surgery status | | | | |
| • Bilateral | 78 (50) | 21 | 23 | 34 |
| • Unilateral | 5 (3) | 1 | 2 | 2 |
| • None | 74 (47) | 32 | 15 | 27 |

AMD = age-related macular degeneration; MGS = Minnesota Grading System.

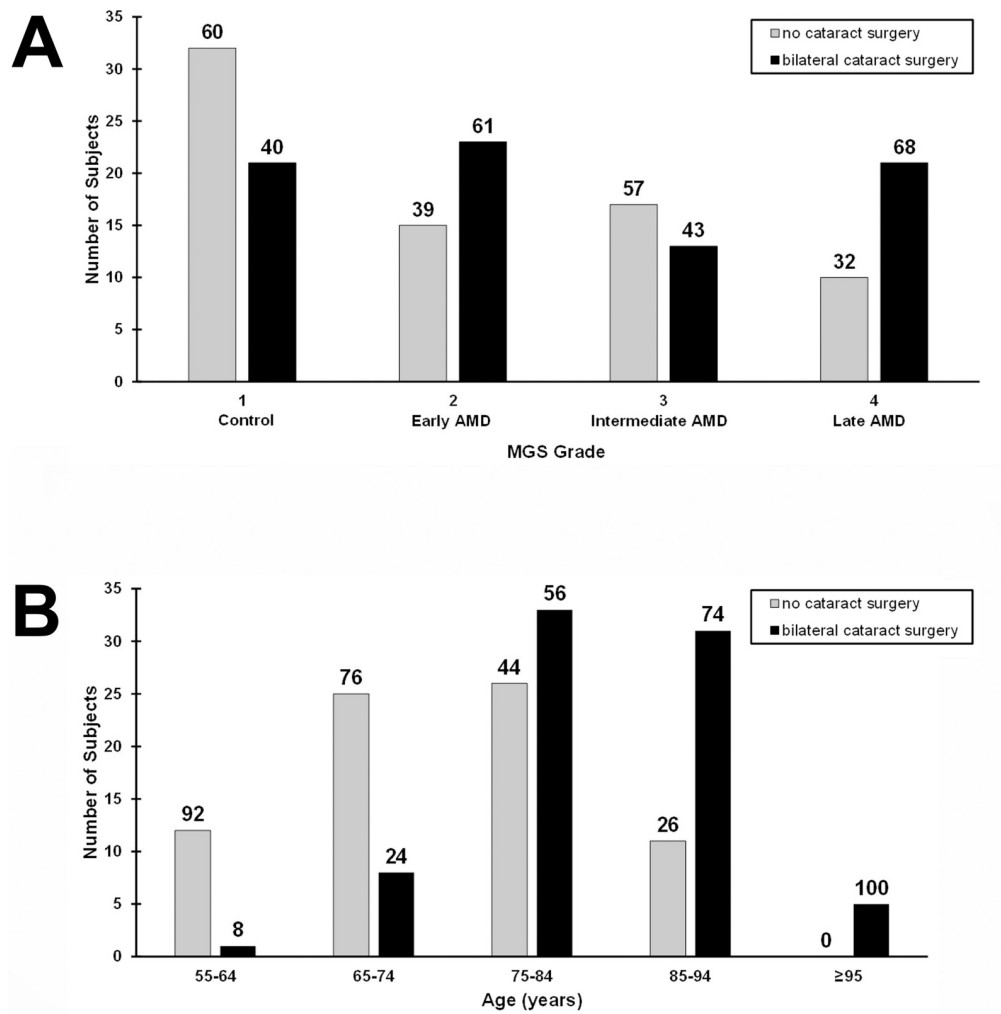

**Fig 1.** Distributions of MGS grade (A) and age (B) in nonsurgical versus bilateral cataract surgery subjects. Absolute numbers are plotted in the histogram plots, and numbers showing the percentage breakdown for each x-axis group appear above each bar. AMD = age-related macular degeneration; MGS = Minnesota Grading System.

To test if cataract surgery might contribute to greater mtDNA damage, we compared subjects without a history of cataract surgery and those with prior bilateral cataract surgery. A boxplot of mean mtDNA lesions for the entire mtDNA genome shows similar distributions in the two groups (Fig 2). Univariable analysis with the Wilcoxon rank sum test verifies no significant difference in mtDNA lesion frequency between those without cataract surgery and those

**Table 2. Ordinal logistic regression analysis of MGS grade, N = 152.**

| Predictor Variable | Coefficient Estimate | Standard Error | P Value | Odds Ratio (95% CI) |
|---|---|---|---|---|
| Cataract surgery | -0.20 | 0.35 | 0.57 | 0.82 (0.42–1.61) |
| Age | 0.090 | 0.021 | < 0.001 | 1.09 (1.05–1.14) |
| Male gender | -0.50 | 0.32 | 0.12 | 0.61 (0.32–1.14) |
| Smoking | -0.15 | 0.31 | 0.64 | 0.86 (0.47–1.59) |

CI = confidence interval; MGS = Minnesota Grading System.

**Table 3. Multiple logistic regression model to predict intermediate/late AMD (MGS 3 and 4, N = 61) versus controls (MGS 1, N = 53).**

| Predictor Variable | Coefficient Estimate | Standard Error | P Value | Odds Ratio (95% CI) |
|---|---|---|---|---|
| Cataract surgery | -0.36 | 0.48 | 0.46 | 0.70 (0.27–1.77) |
| Age | 0.12 | 0.031 | < 0.001 | 1.13 (1.06–1.20) |
| Male gender | -0.64 | 0.45 | 0.16 | 0.53 (0.22–1.28) |
| Smoking | -0.27 | 0.44 | 0.54 | 0.76 (0.32–1.83) |

AMD = age-related macular degeneration; CI = confidence interval; MGS = Minnesota Grading System.

with bilateral cataract surgery ($P$ = 0.49, difference in location -0.050 with 95% confidence interval of -0.199 to 0.107). This narrow confidence interval about the null value provides evidence for sufficient study power supporting the null hypothesis that mean mtDNA lesion frequency does not differ based on history of cataract surgery. Equivalence hypothesis testing also provides evidence for a true lack of association between history of cataract surgery and RPE mtDNA damage in the donor eyes: performing the two-one-sided t-tests procedure on our study data shows a significant result ($P$ = 0.02) with the smallest effect size of interest set at 0.2 and the alpha level set at 0.05, not assuming equal variance. This result allows us to reject a difference in mean RPE mtDNA damage of 0.2 units or greater, corresponding to a difference in group means of 17% or greater, between the donor eyes with and without cataract surgery. Expanding the analysis to a multiple regression model with mean mtDNA lesions as the

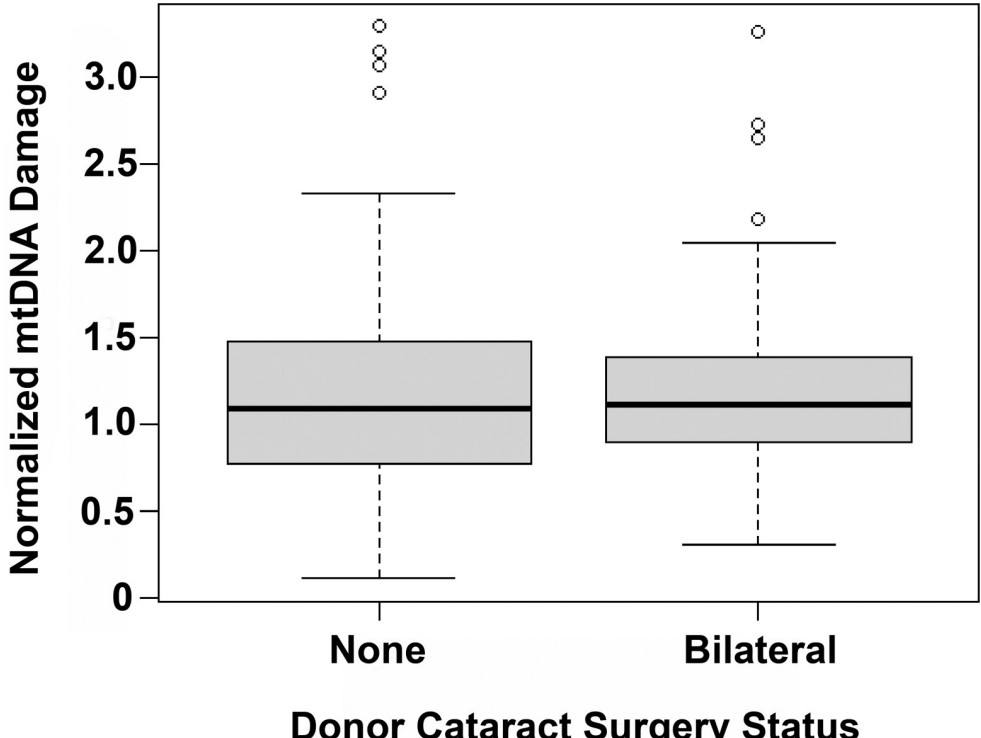

**Fig 2. Normalized RPE mtDNA damage is similar in nonsurgical and bilateral cataract surgery subjects.** The normalized RPE mtDNA damage was calculated for each subject by dividing that subject's mtDNA lesion frequency by the mean mtDNA lesion frequency in the control MGS 1 group. Boxplots show similar distributions of normalized RPE mtDNA damage in subjects without cataract surgery and those with bilateral cataract surgery. MGS = Minnesota Grading System; mt = mitochondrial; RPE = retinal pigment epithelium.

**Table 4. Multiple linear regression analysis of mitochondrial DNA lesion frequency, *N* = 152.**

| Predictor Variable | Coefficient Estimate (95% CI) | Standard Error | *P* Value |
|---|---|---|---|
| Cataract Surgery | -0.048 (-0.27–0.17) | 0.11 | 0.66 |
| Age | 0.0066 (-0.0057–0.019) | 0.0062 | 0.29 |
| Male gender | -0.037 (-0.24–0.17) | 0.10 | 0.72 |
| Smoking | -0.13 (-0.33–0.066) | 0.099 | 0.19 |
| Intercept | 0.79 (-0.17–1.75) | 0.49 | 0.11 |

CI = confidence interval.

dependent variable and cataract surgery status, age, gender, and smoking as the independent variables shows no statistically significant association between mean mtDNA lesions and any of the independent variables in the study population (Table 4).

## Discussion

Our study results show that cataract surgery status is not a significant predictor of MGS grade in human donor eyes after adjusting for covariates including age. Although univariable analysis showed an association between cataract surgery and more advanced AMD based on higher MGS grade in the donor eyes, age was a confounding variable. Advanced age is the strongest risk factor for AMD development [26] and also is a well-established risk factor for cataract formation [27, 28]. Therefore, it is not surprising that our data analysis shows significant associations between (1) age and MGS grade as well as (2) age and bilateral cataract surgery.

Our study showed a lack of association between cataract surgery status and RPE mtDNA damage with both univariable and multivariable analyses. Our prior research found that aging in the absence of AMD was accompanied by selective mtDNA damage that was limited to the mitochondrial region known as the common deletion [17]. In contrast, donors with AMD exhibited mtDNA damage in all regions of the mitochondrial genome. Therefore, even though cataract is associated with aging, extensive RPE mtDNA damage does not occur with aging alone, unless associated with advanced AMD. Similarly, we can assume that changes after cataract surgery would not be associated with increased RPE mtDNA damage unless also associated with more advanced AMD. In this study, since the mtDNA analysis included all regions of the mtDNA genome for each donor eye, our finding that mtDNA damage is similar between the bilateral cataract surgery and no cataract surgery groups provides additional evidence against the hypothesis that cataract surgery accelerates AMD progression.

The effect of cigarette smoking on the risk of developing AMD has been confirmed in several studies. Cigarette smoking has been linked to increased oxidative stress, platelet aggregation, reduction in high-density lipoprotein levels and increased level of multiple inflammatory markers including C-reactive protein, interleukin-6, and tumor necrosis factor alpha [29]. In the US twin study of AMD [30], Seddon and coworkers compared 222 pairs of male twins with intermediate or late-stage AMD to 459 twins with no maculopathy and found that current smokers had a 1.9 times increased risk of AMD while past cigarette smokers had a 1.7 times increased risk when compared to non-smokers. In addition, the Nurses' Health Study demonstrated a dose-response effect of pack-years of smoking that persisted for years after smoking cessation, with nurses who smoked 25 or more cigarettes daily having an increased relative risk of 2.4 [31]. Although our study did not identify an association between smoking status and MGS grade or smoking status and RPE mtDNA damage, we caution about this finding because the medical and ocular history provided in the donor eye report by family members did not quantitate cigarettes per day, packs per year, or years of smoking.

Our study did not find gender to be a significant predictor for MGS grade or RPE mtDNA damage. Female gender is variably linked to AMD in population-based studies. For instance, both the Beaver Dam study [32] and an evaluation of Medicare beneficiaries [33] showed that older women were more likely than older men to have exudative AMD after adjusting for age. However, other studies such as an analysis of the 2005–2008 National Health and Nutrition Examination Survey [34] and a meta-analysis of the European Eye Epidemiology consortium [35] did not find an association between gender and AMD.

Limitations of this study include its retrospective nature, the potentially inaccurate information provided by family members about the donors, and the unclear timing of cataract surgery and AMD diagnosis. Also, clinical ophthalmologic details not evident on donor tissue examination–such as visual acuity, history of exudation secondary to AMD, history of treatment with intravitreal anti-vascular endothelial growth factor injection, and type of intraocular lens implanted–were not available for analysis.

In conclusion, in this study of human donor eyes, no evidence was found for an association between cataract surgery status and RPE mtDNA damage previously associated with advanced AMD. Additionally, no evidence was found for an association between cataract surgery status and AMD severity after adjusting for age. These findings support the hypothesis that cataract surgery does not hasten AMD progression. Although the long-term outcomes of cataract surgery in the setting of AMD are unknown [14], our study adds pathologic and molecular findings to the growing body of clinical evidence supporting the idea that cataract surgery is safe in the setting of AMD.

## Supporting information

**S1 Data. Dataset for this study.**
(XLSX)

## Acknowledgments

The Lions Gift of Sight (formerly known as the Minnesota Lions Eye Bank, St. Paul, MN, URL: https://lionsgiftofsight.umn.edu) supplied donor tissue for this study. We acknowledge Dr. Natnaree Taechajongjintana for her assistance with data collection and Dr. Frederick L. Ferris III for helpful discussion regarding experimental design.

## Author Contributions

**Conceptualization:** Deborah A. Ferrington, Sandra R. Montezuma.

**Data curation:** Karen R. Armbrust, Pabalu P. Karunadharma, Marcia R. Terluk, Rebecca J. Kapphahn, Timothy W. Olsen, Deborah A. Ferrington, Sandra R. Montezuma.

**Formal analysis:** Karen R. Armbrust.

**Funding acquisition:** Deborah A. Ferrington, Sandra R. Montezuma.

**Investigation:** Karen R. Armbrust, Pabalu P. Karunadharma, Marcia R. Terluk, Timothy W. Olsen, Deborah A. Ferrington, Sandra R. Montezuma.

**Methodology:** Karen R. Armbrust, Timothy W. Olsen, Deborah A. Ferrington, Sandra R. Montezuma.

**Supervision:** Deborah A. Ferrington, Sandra R. Montezuma.

**Writing – original draft:** Karen R. Armbrust.

**Writing – review & editing:** Pabalu P. Karunadharma, Marcia R. Terluk, Rebecca J. Kapphahn, Timothy W. Olsen, Deborah A. Ferrington, Sandra R. Montezuma.

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
