## [Decision Letter · Decision Letter 0]

13 Jul 2021

PONE-D-21-15720

No association between cataract surgery and age-related macular degeneration in human donor eyes

PLOS ONE

Dear Dr. Montezuma,

Thank you for submitting your manuscript to PLOS ONE. After careful consideration, we feel that it has merit but does not fully meet PLOS ONE’s publication criteria as it currently stands. Therefore, we invite you to submit a revised version of the manuscript that addresses the points raised during the review process.

The subject is of great interest but there are Several limitations noted by one of the reviewers that need to be addressed. What is the power of this study. It may be that the  study may take the form of a more descriptive style rather than a conclusive one on this subject. 

We look forward to receiving your revised manuscript.

Kind regards,

Demetrios G. Vavvas

Academic Editor

PLOS ONE

Journal Requirements:

2. In your ethics statement in the manuscript methods and the online submission form, please include the full name and URL of the biobank from which samples were obtained.

[I have read the journal's policy and the authors of this manuscript have the following competing interests:

TWO: Equity owner – iMacular Regeneration LLC.

DAF: Scientific advisory board member – Vinci Pharmaceuticals.

The authors have no proprietary or commercial interest in any materials discussed in this article.].

Reviewers' comments:

Reviewer's Responses to Questions

**Comments to the Author**

1. Is the manuscript technically sound, and do the data support the conclusions?

Reviewer #1: No

Reviewer #2: Yes

2. Has the statistical analysis been performed appropriately and rigorously? 

Reviewer #1: Yes

Reviewer #2: Yes

3. Have the authors made all data underlying the findings in their manuscript fully available?

Reviewer #1: Yes

Reviewer #2: Yes

4. Is the manuscript presented in an intelligible fashion and written in standard English?

Reviewer #1: Yes

Reviewer #2: Yes

5. Review Comments to the Author

Reviewer #1: Please see attached comments.

Reviewer #2: The manuscript is well-presented, in standard academic English, and it addresses the topic very well. The authors mention the limitations related to their data collection process, and the statistical analysis has been appropriately performed for the purpose of the study.

6. PLOS authors have the option to publish the peer review history of their article (what does this mean?). If published, this will include your full peer review and any attached files.

Reviewer #1: No

Reviewer #2: **Yes: **Dimitrios Ntentakis

---

## [Author Response · Author response to Decision Letter 0]

26 Aug 2021

Response to Reviewers (the text below also is attached to this revision as a word document labeled 'Response to Reviewers'):

Reviewer #1:

1. They need to use donor eyes only for mtDNA lesion frequency. Other data did not necessarily require the use of human donor eyes. The main conclusion is supposed to be that there is no association between cataract surgery status and RPE mtDNA damage.

Authors’ Response: We agree with the reviewer that the mtDNA lesion frequency analysis requires donor eyes, while methods not requiring human donor eyes can test whether cataract surgery is associated with AMD severity. We have retained the Minnesota Grading System (MGS) pathologic analysis in our manuscript but have incorporated changes to emphasize the conclusion from the mtDNA lesion frequency analysis and specify our study type:

(1) We changed the title to include mitochondrial DNA damage. The original title was “No association between cataract surgery and age-related macular degeneration in human donor eyes” and the new title is “No association between cataract surgery and mitochondrial DNA damage with age-related macular degeneration in human donor eyes.”

(2) In the abstract conclusions section we now specify that our findings are “pathologic and molecular” (page 3, line 44) to differentiate our study from other types of studies.

(3) We added more information to the introduction to specifically describe how using human donor eyes provides novel information. We inserted the following at the end of the introduction: “Inclusion of mtDNA damage analysis from donor eyes provides new information about features of the RPE that are not accessible in living patients due to the invasive nature of measuring damage in the RPE. Therefore, this study extends information provided by previous clinical studies by including a biochemical measure of mtDNA damage in the cell type affected by AMD.” (page 4, lines 77-81; tracked changes lines 78-82)

(4) We have reordered the last paragraph in the Discussion (page 14, lines 260-263; tracked changes lines 263-266) so that the lack of association between cataract surgery status and mtDNA damage is listed first, as the main conclusion. This section originally stated, “In conclusion, in this study of human donor eyes, no evidence was found for an association between cataract surgery status and AMD severity after adjusting for age. Additionally, no evidence was found for an association between cataract surgery status and RPE mtDNA damage.” but now reads, “In conclusion, in this study of human donor eyes, no evidence was found for an association between cataract surgery status and RPE mtDNA damage previously associated with advanced AMD. Additionally, no evidence was found for an association between cataract surgery status and AMD severity after adjusting for age.” Similarly, we reordered the abstract conclusions section (page 3 lines 42-43) so our conclusion about retinal pigment epithelium mitochondrial DNA damage appears first.

2. P5 L98; As previously described [21]...This should be described precisely. It seems there is no new data obtained for this study. They collected mtDNA data from three previous studies; Karunadharma et al. 2010, Terluk et al. 2015, and Exp Eye Res. 2016 (21).

Authors’ Response: Although previously published studies contain the donor eye mtDNA lesion frequency data, this study does contain other new data which then allows for novel analysis. Cataract surgery was not evaluated in any of the prior studies, so the new cataract surgery data allow us to assess for an association between mtDNA damage and history of cataract surgery. We have inserted the statement “We utilized the RPE mtDNA data from this prior study with new cataract surgery data to investigate the potential relationship between RPE mtDNA damage and cataract surgery” in the methods section (page 6, lines 114-116; tracked changes lines 115-117) to clarify which data are new. With the new cataract surgery data, we also perform novel analysis of the relationship between MGS stage (which is based on the Age-Related Eye Disease Study Severity Scale) and cataract surgery status, which corresponds to the type of analysis typically performed in human clinical studies.

3. Methods: Limited description on mtDNA damage and lesion frequency. Not for broad readers. Please explain more about mtDNA damage experiments. Example; qPCR assay permits monitoring the mtDNA integrity directly from total cellular DNA without the need for isolating mitochondria. Lesions such as oxidative DNA damage can slow down DNA polymerase reaction. PCR will amplify normal DNA to a greater extent than damaged DNA under identical conditions. Hence, by comparing PCR amplification efficiency, DNA damage can be expressed in terms of lesions per kilobase mathematically (PMID: 11020328).

Authors’ Response: We thank the reviewer for this suggestion. We have incorporated the suggested reference and added a more detailed description of the mtDNA damage assay to page 5, lines 106-111 (tracked changes lines 107-112), where we inserted the following statement: “This assay uses mitochondrial-specific primers that preferentially amplify mtDNA from total DNA isolated from RPE. The advantage of this method is that mitochondrial purification is not required. Genomic regions without DNA damage amplify more efficiently with quantitative PCR than regions with DNA damage [21], so this assay allows quantitative determination of mtDNA damage based on PCR amplification efficiency.”

4. P4 L69; In support of this hypothesis...Should be described more precisely. Previously a significant decrease in number and area of mitochondria as well as loss of cristae and matrix density were found in AMD and normal aging.

Authors’ Response: We agree with the reviewer that more description of this prior study would be helpful for the reader. This description on Page 4 lines 69-72 (tracked changes lines 70-73) initially read “In support of this hypothesis, a prior study shows decreased number and size of RPE mitochondria in AMD [15]…” and now reads “The link between mtDNA damage and AMD is supported by a prior study that showed decreased number and size of RPE mitochondria as well as loss of RPE mt matrix density and cristae in AMD [15]. Although these changes also occurred in normal aging, the changes were accelerated in AMD.”

5. P4 L69; Should be “mitochondrial (mt) DNA damage”?

Authors’ Response: We agree with the reviewer that the phrase “mitochondrial (mt) DNA damage” is a more accurate description of our study hypothesis than “mitochondrial (mt) damage”, so page 4 line 69 (tracked changes line 70) now states “mitochondrial (mt) DNA damage.”

Reviewer #2:

6. In line 158 of the results section, the authors could consider replacing “lens status” with “history of cataract surgery”. This substitution might better represent the prognostic relevance of intra-operative light exposure and ensuing inflammation that the authors are addressing with their analysis, rather than the lens status itself.

Authors’ Response: We agree with the reviewer’s suggestion, so this phrase, now lines 174-175 (tracked changes line 176), now states “…we conclude that the donor eye history of cataract surgery is not associated with MGS grade.”

Editor:

7. The subject is of great interest but there are several limitations noted by one of the reviewers that need to be addressed. What is the power of this study? It may be that the study may take the form of a more descriptive style rather than a conclusive one on this subject.

Authors’ Response: We appreciate the reviewers’ comments, which we have addressed above. We understand the request for study power, since a null result (i.e. no significant difference in RPE mtDNA lesion frequency between those without cataract surgery and those with bilateral cataract surgery) may arise from a study without enough power to reject the null hypothesis instead of a true lack of effect. We have included additional quantitative data to allay this concern.

 Since post-hoc power analysis using study data merely reports p-values in a different way (Hoenig and Heisey 2001, Levine and Ensom 2001), with high p-values corresponding to low power, we did not conduct an observed post-hoc power analysis. More informatively, a theoretic a priori power analysis shows that a study with the same sample size and standard deviation as the current study would have 80% power at detecting a difference in RPE mtDNA lesion frequency between those without cataract surgery and those with bilateral cataract surgery if the true group means were different by 0.3 units, or 0.3/1.2 = 25%, which is a relatively small difference. {The parameters of the non-surgery group are as follows: n = 74, mean = 1.19, standard deviation = 0.65; the parameters of the history of cataract surgery group are as follows: n = 78, mean = 1.20, standard deviation = 0.52. A two-sample power calculation for a two-sided t-test with n = 74 per group (the smaller n of our two groups), standard deviation = 0.65 (the larger standard deviation of our two groups), difference in means = 0.3, and significance level = 0.05 gives a power of 80%.} The result of this theoretic power calculation provides evidence against an underpowered study, and we go on to provide two quantitative analyses in the manuscript to address the degree of confidence in accepting the null hypothesis of no difference in RPE mtDNA lesion frequency between those without cataract surgery and those with bilateral cataract surgery.

 (1) We have added the difference in location of mean RPE mtDNA lesion frequency for the 2 groups and the corresponding confidence interval to show the range of effect that our data support. The narrow confidence interval about the null supports a true lack of association. The results section (page 10 lines 185-190; tracked changes lines 186-191), which previously stated “Univariate analysis with the Wilcoxon rank sum test verifies no significant difference in mtDNA lesion frequency between those without cataract surgery and those with bilateral cataract surgery (P = 0.49)” now reads, “Univariate analysis with the Wilcoxon rank sum test verifies no significant difference in mtDNA lesion frequency between those without cataract surgery and those with bilateral cataract surgery (P = 0.49, difference in location -0.050 with 95% confidence interval of -0.199 to 0.107). This narrow confidence interval about the null value provides evidence for sufficient study power supporting the null hypothesis that mean mtDNA lesion frequency does not differ based on history of cataract surgery”

 (2) Alternatively, to assess the lack of association between history of cataract surgery and mean RPE mtDNA lesion frequency, we performed equivalence hypothesis testing to test whether we can reject an effect at least as extreme as the smallest effect size of interest, which in this case is a specified difference in the mean RPE mtDNA damage values between the donor eyes with and without cataract surgery. To determine whether we can reject the small difference in the mean RPE mtDNA lesion frequency of 0.2 units or greater (0.2/1.2 = 17% difference in means) between the groups with and without cataract surgery, we performed the two-one-sided t-tests procedure with our study data, setting the smallest effect size of interest as 0.2 and the alpha level as 0.05, and not assuming equal variance. This equivalence test was significant (P = 0.02), which rejects the presence of a difference in means of 0.2 or greater and indicates that the observed effect in our study between the two groups is statistically equivalent at this level. We have added the equivalence hypothesis testing statistical package to the methods (page 6 line 124; tracked changes line 125) and the equivalence hypothesis testing results to the manuscript on page 10 lines 190-196 (tracked changes lines 191-197), where we have inserted the following result: “Equivalence hypothesis testing also provides evidence for a true lack of association between history of cataract surgery and RPE mtDNA damage in the donor eyes: performing the two-one-sided t-tests procedure on our study data shows a significant result (P = 0.02) with the smallest effect size of interest set at 0.2 and the alpha level set at 0.05, not assuming equal variance. This result allows us to reject a difference in mean RPE mtDNA damage of 0.2 units or greater, corresponding to a difference in group means of 17% or greater, between the donor eyes with and without cataract surgery.”

8. Please ensure that your manuscript meets PLOS ONE's style requirements, including those for file naming. The PLOS ONE style templates can be found at https://journals.plos.org/plosone/s/file?id=wjVg/PLOSOne_formatting_sample_main_body.pdf and https://journals.plos.org/plosone/s/file?id=ba62/PLOSOne_formatting_sample_title_authors_affiliations.pdf

Authors’ Response: We confirm that our manuscript and file naming meet PLOS ONE’s style requirements.

9. In your ethics statement in the manuscript methods and the online submission form, please include the full name and URL of the biobank from which samples were obtained.

 Authors’ Response: We have added the URL to the manuscript methods (page 4 line 86) as well as the acknowledgments section, so now the full name and URL of the biobank from which the samples were obtained appear in the manuscript. We also added the full name and URL of the biobank to the Ethics Statement field of the online submission, where we added the following statement: “The samples used in this study were human donor eyes obtained from the Minnesota Lions Eye Bank (renamed the Lions Gift of Sight in 2018, St. Paul, MN, URL: https://lionsgiftofsight.umn.edu).”

10. PLOS requires an ORCID iD for the corresponding author in Editorial Manager on papers submitted after December 6th, 2016. Please ensure that you have an ORCID iD and that it is validated in Editorial Manager. To do this, go to ‘Update my Information’ (in the upper left-hand corner of the main menu), and click on the Fetch/Validate link next to the ORCID field. This will take you to the ORCID site and allow you to create a new iD or authenticate a pre-existing iD in Editorial Manager. Please see the following video for instructions on linking an ORCID iD to your Editorial Manager account: https://www.youtube.com/watch?v=_xcclfuvtxQ

Authors’ Response: Dr. Sandra Montezuma, the corresponding author, has linked her ORCID iD to her account in Editorial Manager.

11. Thank you for stating the following in the Competing Interests section: [I have read the journal's policy and the authors of this manuscript have the following competing interests: TWO: Equity owner – iMacular Regeneration LLC. DAF: Scientific advisory board member – Vinci Pharmaceuticals. The authors have no proprietary or commercial interest in any materials discussed in this article.]. Please confirm that this does not alter your adherence to all PLOS ONE policies on sharing data and materials, by including the following statement: "This does not alter our adherence to PLOS ONE policies on sharing data and materials.” (as detailed online in our guide for authors http://journals.plos.org/plosone/s/competing-interests). If there are restrictions on sharing of data and/or materials, please state these. Please note that we cannot proceed with consideration of your article until this information has been declared. Please include your updated Competing Interests statement in your cover letter; we will change the online submission form on your behalf.

Authors’ Response: We confirm that these competing interests do not alter our adherence to PLOS ONE policies on sharing data and materials. We have updated our Competing Interests statements in the cover letter accordingly. Thank you for updating the information in the online submission form. We confirm that all data are fully available without restriction.

References:

Hoenig JM, Heisey DM. The abuse of power: the pervasive fallacy of power calculations for data analysis. Am Stat. 2001;55:19-24.

Levine M, Ensom MH. Post hoc power analysis: an idea whose time has passed? Pharmacotherapy. 2001 Apr;21(4):405-9.

---

## [Decision Letter · Decision Letter 1]

6 Oct 2021

No association between cataract surgery and mitochondrial DNA damage with age-related macular degeneration in human donor eyes

PONE-D-21-15720R1

Dear Dr. Montezuma,

We’re pleased to inform you that your manuscript has been judged scientifically suitable for publication and will be formally accepted for publication once it meets all outstanding technical requirements.

Kind regards,

Demetrios G. Vavvas

Academic Editor

PLOS ONE

Additional Editor Comments (optional):

Reviewers' comments:

Reviewer's Responses to Questions

**Comments to the Author**

1. If the authors have adequately addressed your comments raised in a previous round of review and you feel that this manuscript is now acceptable for publication, you may indicate that here to bypass the “Comments to the Author” section, enter your conflict of interest statement in the “Confidential to Editor” section, and submit your "Accept" recommendation.

Reviewer #1: All comments have been addressed

2. Is the manuscript technically sound, and do the data support the conclusions?

Reviewer #1: Yes

3. Has the statistical analysis been performed appropriately and rigorously? 

Reviewer #1: Yes

4. Have the authors made all data underlying the findings in their manuscript fully available?

Reviewer #1: Yes

5. Is the manuscript presented in an intelligible fashion and written in standard English?

Reviewer #1: Yes

6. Review Comments to the Author

Reviewer #1: (No Response)

7. PLOS authors have the option to publish the peer review history of their article (what does this mean?). If published, this will include your full peer review and any attached files.

Reviewer #1: **Yes: **Shoji Notomi

---

## [Editor Report · Acceptance letter]

11 Oct 2021

PONE-D-21-15720R1 

No association between cataract surgery and mitochondrial DNA damage with age-related macular degeneration in human donor eyes 

Dear Dr. Montezuma:

I'm pleased to inform you that your manuscript has been deemed suitable for publication in PLOS ONE. Congratulations! Your manuscript is now with our production department. 

Kind regards, 

on behalf of

Prof. Demetrios G. Vavvas 

Academic Editor

PLOS ONE